# Exploring Natural Alkaloids from Brazilian Biodiversity as Potential Inhibitors of the *Aedes aegypti* Juvenile Hormone Enzyme: A Computational Approach for Vector Mosquito Control

**DOI:** 10.3390/molecules28196871

**Published:** 2023-09-29

**Authors:** Renato Araújo da Costa, Andréia do Socorro Silva da Costa, João Augusto Pereira da Rocha, Marlon Ramires da Costa Lima, Elaine Cristina Medeiros da Rocha, Fabiana Cristina de Araújo Nascimento, Anderson José Baia Gomes, José de Arimatéia Rodrigues do Rego, Davi do Socorro Barros Brasil

**Affiliations:** 1Laboratory of Biosolutions and Bioplastics of the Amazon, Graduate Program in Science and Environment, Institute of Exact and Natural Sciences, Federal University of Pará (UFPA), Belém 66075-110, PA, Brazil; ac165051@gmail.com (A.d.S.S.d.C.); fabiananascimento987@gmail.com (F.C.d.A.N.); jr2rego@gmail.com (J.d.A.R.d.R.); davibb@ufpa.br (D.d.S.B.B.); 2Laboratory of Molecular Biology, Evolution and Microbiology, Federal Institute of Education, Science and Technology of Pará (IFPA) Campus Abaetetuba, Abaetetuba 68440-000, PA, Brazil; ramireslima9774@gmail.com (M.R.d.C.L.); anderson.gomes@ifpa.edu.br (A.J.B.G.); 3Graduate Program in Chemistry, Federal University of Pará (UFPA), Belém 66075-110, PA, Brazil; joao.rocha@ifpa.edu.br (J.A.P.d.R.); elaine.rocha@ifpa.edu.br (E.C.M.d.R.); 4Laboratory of Modeling and Computational Chemistry, Federal Institute of Education, Science and Technology of Pará (IFPA) Campus Bragança, Bragança 68600-000, PA, Brazil

**Keywords:** insecticide, *Aedes aegypti* mosquito, juvenile hormone, molecular dynamics, free energy

## Abstract

This study explores the potential inhibitory activity of alkaloids, a class of natural compounds isolated from Brazilian biodiversity, against the mJHBP enzyme of the *Aedes aegypti* mosquito. This mosquito is a significant vector of diseases such as dengue, zika, and chikungunya. The interactions between the ligands and the enzyme at the molecular level were evaluated using computational techniques such as molecular docking, molecular dynamics (MD), and molecular mechanics with generalized Born surface area (MMGBSA) free energy calculation. The findings suggest that these compounds exhibit a high binding affinity with the enzyme, as confirmed by the binding free energies obtained in the simulation. Furthermore, the specific enzyme residues that contribute the most to the stability of the complex with the compounds were identified: specifically, Tyr33, Trp53, Tyr64, and Tyr129. Notably, Tyr129 residues were previously identified as crucial in the enzyme inhibition process. This observation underscores the significance of the research findings and the potential of the evaluated compounds as natural insecticides against *Aedes aegypti* mosquitoes. These results could stimulate the development of new vector control agents that are more efficient and environmentally friendly.

## 1. Introduction

The World Health Organization reports that vector-borne diseases account for over 17% of all infectious diseases, causing more than 700,000 deaths annually [1]. Approximately 80% of the global population is at risk of contracting these diseases, making them a persistent public health issue [2]. Diseases such as dengue, zika, yellow fever, and chikungunya, primarily transmitted by the *Aedes aegypti* mosquito, exacerbate this concern [3,4]. These mosquitoes possess an inherent ability to adapt and proliferate, contributing to their widespread distribution and effective disease transmission [5,6,7]. Urbanization has facilitated the expansion of environments conducive to vector growth, while rapid global migrations have amplified the activation potential of vectors and viral agents. Resource scarcity, often exacerbated by poverty, impedes the ability of individuals and communities to implement effective protective measures. Moreover, even when control resources are accessible, their application frequently lacks efficacy. Effective management of the *Aedes aegypti* mosquito is crucial in curbing disease spread, with historical instances demonstrating successful disease eradication or significant reduction through vector control [8,9]. Given the limited treatment options for these diseases, controlling the mosquito population that transmits them is a viable strategy to reduce their spread. Various strategies have been proposed for mosquito vector control [10]. The most prevalent method involves the use of chemical insecticides, many of which have proven effective against *Aedes aegypti* larvae and adults, including carbamates, pyrethroids, and organophosphates. However, these chemical insecticides are non-selective and can harm the environment [2,11,12,13,14,15,16,17].

Natural products (PCNs) serve as a rich reservoir of secondary metabolites, offering a promising alternative for mosquito control [2,18,19]. They exhibit lower toxicity, high efficacy in minimal quantities, biodegradability, and less inducement of insect resistance, making them superior alternatives to synthetic compounds [20,21,22,23]. Research has demonstrated the effectiveness of plant-based bioinsecticides in controlling mosquitoes, such as *Aedes aegypti*, that pose significant public health risks [2,16,22,24,25,26,27,28,29].

Alkaloids, a notable category of secondary metabolites, exhibit numerous bioinsecticidal properties [30,31,32,33,34,35,36,37,38]. These compounds manipulate redox reactions, regulate hormonal activity, modify neural impulses, and disrupt cellular and physiological functions, thereby obstructing all biological processes and hindering reproduction [39]. Consequently, biopesticides derived from alkaloids could serve as an alternative or supplementary approach for enhancing mosquito control products.

To deal with *Aedes aegypti*’s resistance to current insecticides, it is rational to design compounds with mechanisms of action different from those currently in use. The biorational approach is a strategy used to design compounds that interact with a specific biological target of the target organism to modulate its activity and impair mosquito activity, causing little or no toxicity or side effects in non-target organisms and the environment. A biorational insecticide interferes with all activities involved in that location, making it more specific and safer than a conventional insecticide. The target is ideally a mosquito-specific protein to minimize toxicity to humans and other animals [19,40].

The juvenile hormone (JH) biosynthetic pathway is a prime target for the development of biorational insecticides. JH, a sesquiterpenoid produced by corpora allata (CA), regulates insect growth, development, metamorphosis, reproduction, differentiation, and mating behavior [41,42,43]. Given its specificity for insects and other arthropods and its absence in vertebrates, the JH pathway is deemed a suitable target for creating new insecticides [42,44].

The mosquito JH binding protein (mJHBP), a member of the odorant-binding protein (OBP) family in insects and related to D proteins, plays a crucial role in transporting JH hormones within insects [44]. Research has been conducted to investigate the JH biosynthetic pathway and develop JH inhibitors (anti-JH). These compounds, also referred to as insect growth regulators (IGRs), mimic the action of JHs. They interfere with an insect’s developmental or reproductive capabilities, reduce its fertility, or inhibit its maturation into an adult [42,45,46].

This study aims to identify and investigate in silico natural alkaloid products with potential anti-mosquito activity from Brazilian biodiversity by targeting JH-mediated pathways through the mJHBP. The methods employed include molecular docking, dynamic simulation, and free energy calculation using the method of molecular mechanics with generalized Born surface area (MMGBSA).

While several approaches exist for scoring schemes, including the analysis of interaction energies between a protein and its ligand, it is important to note that current methodologies lack sufficient accuracy to reliably predict both the native binding mode and the associated free energy. This limitation stems directly from the need to strike a balance between accuracy and efficiency when screening large compound libraries. As a result, evaluation functions often quantify a simplified representation of the complex protein–ligand interaction. They focus primarily on highly relevant elements, such as hydrogen bonds and hydrophobic interactions, while overlooking aspects like polarization and entropy. To decrease the computational time required in virtual screening, a simplified scoring function is frequently used, which typically considers a limited number of degrees of freedom. Consequently, our research group aims to validate the presented results with in vitro tests conducted through partnerships [47,48,49,50].

## 2. Results and Discussion

### 2.1. Molecular Docking

Prior to docking the alkaloids examined in this study, the docking method underwent validation through re-docking. The cocrystallised JH3 ligand was repositioned into the mJHBP protein binding site. The re-docking of JH3 into the mJHBP receptor yielded an RMSD value of 0.26 Å between the pose achieved in the docking and the experimental pose. This result suggests that the GOLD program’s fitting procedure exhibits satisfactory accuracy in repositioning JH3 within the active site of mJHBP. This is because, as stated by Wang et al. [51], a prediction is deemed successful when the RMSD between the embedded binding pose and the crystallographic ligand’s pose is 2.0 Å or less (Figure 1). Figure 1 illustrates the superimposed alignment of the re-docked pose and the crystallographic ligand.

Following validation, 221 alkaloids from the NuBBEDB database underwent virtual screening. The three compounds with the highest energy, as determined by the PLPchem score function from the GOLD program, were selected. These were used to assess the interactions of these structures within the active site of the mJHBP protein. These interactions were then compared with those of the crystallographic ligand JH3 and the insecticide pyriproxyfen. The results for ligand docking with mJHBP are presented in Table 1 and Figure 2. Appendix A, found in the Appendix A, provides the codes and interaction energies for all NuBBEDB alkaloids used in this study.

Based on the affinity energies obtained by the ChemPLP scoring function, the three alkaloids, NuBBE_1105, NuBBE_1106, and NuBBE_1107, exhibited higher affinity for the active site of the mJHBP protein than for JH3 and pyriproxyfen (Table 1). This suggests that these compounds could potentially serve as mJHBP inhibitors. Figure 2 illustrates the interactions of these selected compounds with the mJHBP active site. The oxygen atom of the pyran ring in each compound forms a hydrogen bond with the phenolic hydroxyl of Tyr129 in the mJHBP active site. This finding aligns with Kim et al.’s findings [44], where the epoxy group of JH3 also forms a hydrogen bond with the phenolic hydroxyl of Tyr129. Additionally, the three compounds form two other hydrogen interactions with the residues Trp53 and Tyr64, suggesting a more effective interaction with the enzyme’s active site and potential insecticidal properties.

The selected compounds were also observed to have hydrophobic interactions with the residues Tyr33, Leu37, Val51, Trp53, Pro55, Val65, Val68, Leu74, Tyr133, Ile140, Phe144, and Ala281, thereby limiting the active site of mJHBP. These findings are consistent with other molecular docking studies targeting the mJHBP enzyme [52,53,54,55].

### 2.2. Molecular Dynamics

Molecular dynamics (MD) simulations serve as a prevalent tool for exploring phenomena in biomolecules, including proteins. This technique is especially effective for examining intricate atomic-level motions, which pose significant challenges for experimental observation [56].

The RMSDs of the apo protein, the three mJHBP–alkaloid complexes (NuBBE_1105, NuBBE_1106 and NuBBE_1107), and the reference complexes (mJHBP–JH3 and mJHBP–pyriproxyfen) were analyzed to determine the deviation of the protein backbone atoms from their initial trajectory positions. This evaluation was crucial in assessing the stability and convergence of the complexes throughout the simulation period.

Figure 3 illustrates that the protein backbone RMSDs for all systems remained stable, with no significant conformational changes throughout the simulation period. The dynamic evolution of the mJHBP backbone structure revealed average RMSD values of 1.54, 1.87, 1.42, 1.34, 1.68, and 1.64 Å for the systems in which the receptor was complexed with NuBBE_1105, NuBBE_1106, NuBBE_1107, JH3, pyriproxyfen, and the Apo protein, respectively. The mJHBP–NuBBE_1107 and mJHBP–JH3 complexes exhibited similar average RMSD values, indicating a comparable conformation.

The mJHBP–NuBBE_1106 complex exhibited the most significant deviation. In comparison to the Apo protein, the alkaloids NuBBE_1105 and NuBBE_1107, when complexed with the mJHBP protein, displayed lower mean deviations. This indicates that the protein is more stable when complexed with these molecules than in its unbound state, which can be attributed to the interactions between these molecules and the protein site’s amino acid residues. The average RMSD values for the compounds NuBE_1105 and NuBBE_1107 were small compared with the insecticide pyriproxyfen (utilized as a standard) in complex with the mJHBP protein, indicating enhanced system stability. This increased stability can be attributed to the interactions between these ligands and the protein’s active site.

The root-mean-square fluctuation (RMSF) values were also calculated to investigate the fluctuations of each atom along the protein chain, which were induced by ligand interactions [57]. This analysis provides insights into the impact of ligand binding on protein flexibility. In the RMSF plots, high fluctuation values indicate highly flexible regions, while lower values denote areas of significant system stability during MD simulations. Figure 4 presents the RMSF plots for both the Apo protein and the protein complexed with the ligands.

Figure 4 presents the RMSF plot for the Apo e protein, complexed with three alkaloids and two reference compounds. The graph reveals that the loop regions exhibit the most significant fluctuations, which naturally occur due to their inherent flexibility. Conversely, the alpha-helices and beta strands, being more rigid than the loop regions, demonstrate the highest stability and, thus, lesser fluctuations.

The Apo mJHBP protein exhibited an average fluctuation of 0.98 Å. In contrast, the mJHBP–NuBBE_1105, mJHBP–NuBBE_1106, and mJHBP–NuBBE_1107 complexes, as well as the mJHBP–JH3 and mJHBP–pyriproxyfen standards, demonstrated fluctuations of 0.88, 0.97, 0.84, 0.81, and 0.96 Å, respectively. These findings suggest that the protein exhibits greater stability in its bound state than in its Apo form. This increased stability can be attributed to the interactions between these ligands and the protein’s active site during MD simulations.

The radius of gyration (Rg) was evaluated to determine the stability of protein–ligand systems based on the compactness of the protein structure during MD simulations [57,58]. The Rgs of both the Apo protein and protein complexes were measured to understand the compactness of the protein during ligand interactions, as depicted in Figure 5. The Rg for the Apo protein was found to be 19.65 Å. In contrast, the observed values for the mJHBP–JH3, mJHBP–pyriproxyfen, mJHBP–NuBBE_1105, mJHBP–NuBBE_1106, and mJHBP–NuBBE_1107 complexes were 19.63, 19.79, 19.73, 19.76, and 19.65 Å, respectively. The findings indicate that the protein is marginally more compact when bound with the JH3 and NuBBE_1107 ligands than with the other ligands. These findings are consistent with the RMSD and RMSF data.

### 2.3. Binding Free Energy Calculations

The MMGBSA method, utilizing MD trajectories, is a prevalent approach for determining binding free energies (Δ*G*_bind_) and thereby assessing the strength of interactions between a ligand and its receptor [59].

Table 2 presents the Δ*G*_bind_ values and the van der Waals (Δ*E*_vdW_), electrostatic (Δ*E*_ele_), polar (Δ*G*_GB_), and nonpolar (Δ*G*_SA_) energy contributions of the three selected alkaloids selected by molecular anchoring (NuBBE_1105, NuBBE_1106, and NuBBE_1107) and the two reference composites used in this study.

The van der Waals (∆*E*_vdW_), electrostatic (∆*E*_ele_), and nonpolar solvation (∆*G*_SA_) energies significantly contributed to the total binding energies of the complexes. Conversely, the polar solvation energy unfavorably impacted the binding energy (Table 2).

The MMGBSA binding free energies for the complexes were ranked as follows: NuBBE_1107 < NuBBE_1106 < JH3 < pyriproxyfen < NuBBE_1105. The alkaloids NuBBE_1105 and NuBBE_1107 exhibited the lowest binding free energy in comparison to the two reference compounds, JH3 and pyriproxyfen. This suggests a superior binding affinity of these alkaloids to the mJHBP enzyme. Consequently, these findings imply that NuBBE_1107 and NuBBE_1105 could be more potent inhibitors of the mJHBP enzyme than JH3 and pyriproxyfen. Literature indicates a linear correlation between the computationally estimated free binding energies and the IC50 values of the compounds [60,61]. The MMGBSA method has been effectively utilized in prior computational studies to estimate and select more potent inhibitors of the mJHBP protein [52,53,55], further supporting our findings.

Notably, the classification of ΔGbind for the three alkaloids (NuBBE_1107, NuBBE_1105, and NuBBE_1106) aligns with the molecular docking results as per the PLPchem scoring function (Table 1).

Protein–ligand interactions are generally stabilized by various types of interactions, with hydrogen bonds being one of the most significant [57]. This study investigated the hydrogen bonds between the mJHBP protein and five ligands over a 100 ns MD simulation. The results are presented in Figure 6 and Table 2.

Figure 6 and Table 3 show that the alkaloids NuBBE_1105 and NuBBE_1107 form an average of three hydrogen bonds. NuBBE_1105 interacted with the residues Trp53, Tyr64, and Tyr129, with occupancies of 6.96%, 95.93%, and 63.50%, respectively. Similarly, NuBBE_1107 interacted with the same residues, with occupancies of 20.62%, 75.05%, and 46.13%, respectively. These figures exceed the interaction numbers observed for the reference compounds, JH3 and pyriproxyfen, which interacted with the residues Tyr33, Tyr129, Tyr64, and Tyr148, with occupancies of 9.26%, 72.47%, 6.62%, and 2.41%, respectively. These data account for the higher affinities of NuBBE_1105 and NuBBE_1107 for the protein, as indicated by the MMGBSA binding free energy values in Table 2. Notably, the interactions of NuBBE_1105 and NuBBE_1107 are conserved from the molecular dock.

To elucidate the specific contribution of each residue to the Δ*G*_bind_, we conducted a decomposition energy analysis of every amino acid residue in the protein using the MMGBSA method. Residues contributing binding free energy values of −1.00 kcal/mol or less were deemed significant for the binding process.

The energy contributions of each residue in the mJHBP–NuBBE_1107 and mJHBP–NuBBE_1105 complexes, which exhibited more favorable binding free energy, were compared with those in the complexes formed by the reference compounds JH3 and pyriproxyfen, as shown in Figure 7.

Figure 7 illustrates a higher count of residues with substantial contributions from the ligands NuBBE_1107 and NuBBE_1105, in comparison with the reference ligands JH3 and pyriproxyfen. This observation explains the enhanced binding affinity of these ligands with the mJHBP enzyme, as indicated in Table 2, thereby validating our findings.

The mJHBP–NuBBE_1107 complex (Figure 7D) is primarily stabilized by the residues Tyr33, Val51, Tpr53, Tyr64, Val65, Val68, Tyr129, Ile140, and Phe144. These residues contribute significantly to the total energy, with respective energy values of −4.13, −1.71, −2.54, −2.40, −1.77, −1.17, −1.30, −1.09, and −1.10 kcal·mol^−1^. In the mJHBP–NuBBE_1105 complex (Figure 7C), the residues Tyr33, Val51, Tpr53, Tyr64, Val65, Val68, Tyr129, and Phe144 contributed significantly to the complex’s stability, exhibiting energy values of −3.82, −1.95, −3.08, −2.94, −1.48, −1.44, −1.10, and −1.05 kcal·mol^−1^, respectively. These findings indicate that the enzyme–NuBBE_1107 complex provides more favorable contributions to the binding free energy than the NuBBE_1105 complex. Residues with relatively large differences in their contributions to the total binding free energies warrant further attention. In the NuBBE_1107 system, the residues Tyr33, Trp53, Tyr64, and Tyr129 are particularly noteworthy, while in the NuBBE_1105 system, residues Tyr33, Trp53, and Tyr64 are most significant.

The residues Tyr33, Trp53, Tyr64, and Tyr129 play a significant role for both ligands. Interactions with the residues Trp53, Tyr64, and Tyr129 within the enzyme’s active site are preserved in molecular docking (Figure 2) and remain consistent during MD simulations. This consistency underscores the critical nature of these interactions for the inhibitory activities of these molecules against the target protein. According to Kim et al. [49] and Kim and collaborators [44], the Tyr129 residue is a pivotal element in the enzyme inhibition process. This residue also interacts with the crystallographic inhibitor JH3, which supports our findings. Our in silico results suggest that the molecules NuBBE_1107 and NuBBE_1105 could potentially inhibit the enzyme mJHBP, thereby serving as potent natural insecticides against *Aedes aegypti* mosquitoes.

## 3. Materials and Methods

### 3.1. Molecular Docking

The RCSB Protein Data Bank (RCSB PDB) (http://www.rcsb.org/ (accessed on 3 April 2023)) was used to obtain the crystal structure of the mosquito juvenile hormone binding protein *Aedes aegypti* (mJHBP) (código PDB: 5V13) [44] and the crystallographic ligand methyl methyl (2E,6E)-9-[(2R)-3,3-dimethyloxiran-2-yl]-3,7-dimethylnona-2,6-dienoate (JH3). The target was prepared by first extracting the water molecules and separating the JH3 from the protein–ligand complex. Hydrogen atoms were added using the H++ server [62]. First, the JH3 ligand was re-fitted to the target to validate molecular docking using the GOLD 2020.1 program (Genetic Optimization for Ligand Docking) [51,63,64]. The pose obtained by the GOLD program was superimposed onto that of the crystallographic ligand, and the mean-square deviation (RMSD) value was obtained. When the RMSD between the embedded binding pose and that of the crystallographic ligand is equal to or less than 2.0 Å, the prediction is considered successful [51]. The ChemPLP (piecewise linear potential) scoring function was used to model the steric complementarity of the protein and the ligand [65]. The binding site was defined as a 10 Å sphere centered on the crystallographic ligand. In order to apply optimal settings for each binder, 100% search efficiency was employed. After validation, 221 alkaloids from the Natural Products Database for the Biodiversity of Brazil (NuBBEDB) [66,67] were subjected to molecular docking following the same redocking protocol. The intermolecular interactions of selected poses were visualized using Discovery Studio.

### 3.2. Molecular Dynamics Simulations

For performing the DM simulation, executed on a graphics processing unit (GPU) in Amber22 [68], the three highest-scoring ligands in molecular docking were used, the co-crystallographic ligand JH3 and pyriproxyfen, used as reference compounds (Table 1).

The structures of the protein and ligands were treated using amber ff14SB and the amber general force field (GAFF), respectively [69,70]. The restricted electrostatic potential (RESP) procedure was used to compute the ligands’ atomic charges using the Gaussian 09 program at the HF/6-31G* level of theory. Using the H++ server, pKa calculations for the ionizable residues in protein structures were used to assess the protonation states of the residues [62]. All systems were solvated in the tLeap module using an octahedral water-box and the TIP3P model [71]. To keep the systems’ electroneutrality, Na+ ions were introduced.

The systems were initially minimized following four steps: (i) ions and water molecules; (ii) hydrogen atoms; (iii) water molecules and hydrogen atoms; and (iv) all systems. All steps were performed using 5000 steps with steepest descent and 5000 more steps with the conjugate gradient algorithm.

Each system was then gradually heated to 300 K during 200 ps; next, 300 ps of density equilibration, with positional restrictions on the protein–ligand atoms, was performed at a constant volume. All protein–ligand complexes were equilibrated with 500 ps of MD without positional restrictions at a steady pressure prior to the production process. By using a collision frequency of 2 cm^−1^ and linking to a Langevin thermostat, the temperature was kept at 300 K. The SHAKE [72] algorithm and the particle mesh Ewald (PME) [73,74] approach were used to limit the bond lengths involving the hydrogen atoms, and a cutoff of 8 was applied for non-bonded interactions. Finally, without applying positional restrictions, 100 ns of time at a 300 K temperature was used for the MD simulations (production). The root-mean-square deviation (RMSD), root-mean-square fluctuation (RMSF), radius of gyration (Rg), and hydrogen bonds were calculated to monitor the stability of the MD simulations.

### 3.3. Binding Free Energy and Residual Decomposition Analysis

The generalized Born surface area (MMGBSA) (Costa, et al., 2022; E. Wang et al., 2019) [57,75] approach was used to calculate the binding free energies (∆*G*_bind_) and the contributions of individual residues by free energy decomposition analysis of each ligand–protein complex using MPBSA.py [76] with the last 10 ns of the MD simulations of each system, according to Equation (1):∆*G*_bind_ = ∆*E*_MM_ + ∆*G*_SOLV_ − *T*∆*S*(1)
where ∆EMM, gas-phase MM energy, is computed with Equation (2):∆E_MM_ = ∆*E*_int_ + ∆E_ele_ − ∆*E*_vdw_(2)
where the changes in internal energies (bond, angle, and dihedral), electrostatic energies, and van der Waals energies, respectively, are denoted by ∆Eint, ∆Eele, and ∆EvdW.

∆G_SOLV_ is the sum of the polar contribution (∆G_GB_) and the nonpolar contribution (∆G_SA_), per Equation (3):∆*G*_SOLV_ = ∆*G_GB_* + ∆*G*_SA_(3)

The solvent-accessible surface area (SASA) approach was used to estimate ∆*G*_SA_ [77]. For estimates of binding free energy, the entropic term (−*T*∆*S*), which has a high computational cost, was disregarded [78].

## 4. Conclusions

In conclusion, this study employed an approach based on the in silico approach to identify and study potential natural compounds from the alkaloid class with the mosquito juvenile hormone binding protein of Aedes aegypti (mJHBP) as candidates for the development of natural insecticides. It identified two compounds, NuBBE_1107 and NuBBE_1105, with promising characteristics and high anchoring energy against mJHBP, capable of forming stable complexes with the protein according to the results of MD simulation. The two compounds showed higher binding energy values than those obtained for the reference ligands (JH3 and pyriproxyfen), demonstrating that the molecules proposed in our study are promising inhibitors of the selected targets. According to the results on energy decomposition per residue, the new inhibitors have a higher interaction profile than that exhibited by the control compounds. These results can be used in the development of new, more effective vector control agents with less environmental impact, and, furthermore, other studies could prioritize and validate these compounds in in vitro studies and in vivo assays.

## Figures and Tables

**Figure 1 molecules-28-06871-f001:**
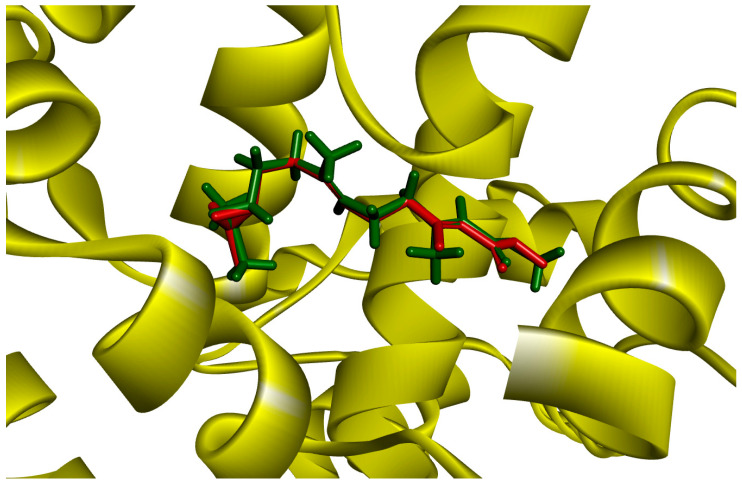
JH3 and mJHBP docking by GOLD (red: crystal structure, green: docking structure).

**Figure 2 molecules-28-06871-f002:**
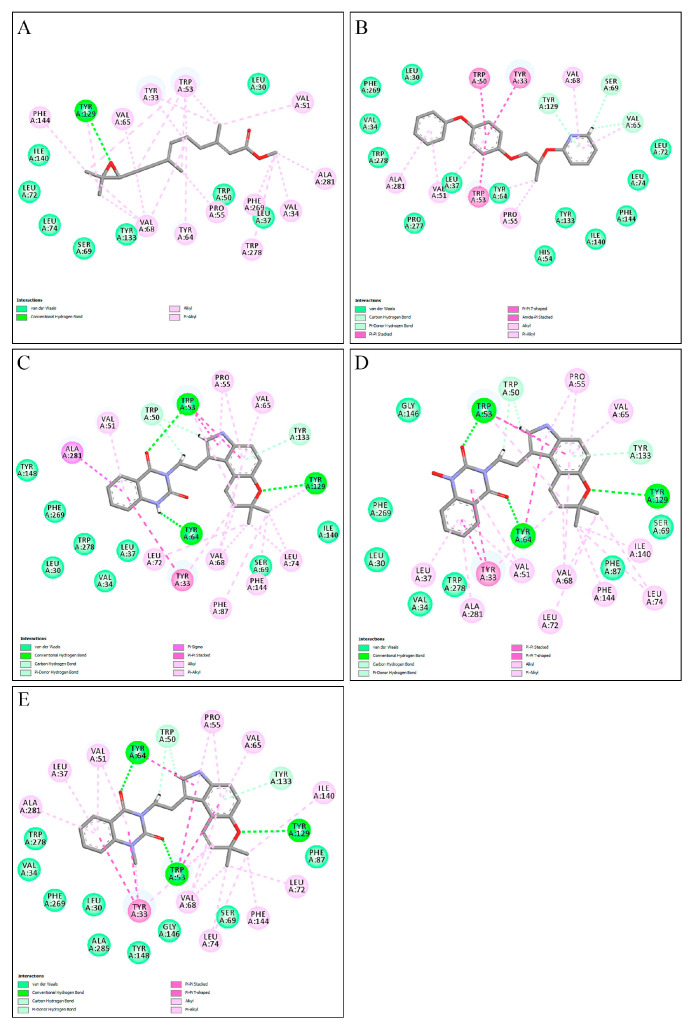
Interactions of JH3 (**A**), pyriproxyfen (**B**), NuBBE_1107 (**C**), NuBBE_1105 (**D**), and NuBBE_1106 (**E**) ligands made with amino acid residues of the active site of the mJHBP protein.

**Figure 3 molecules-28-06871-f003:**
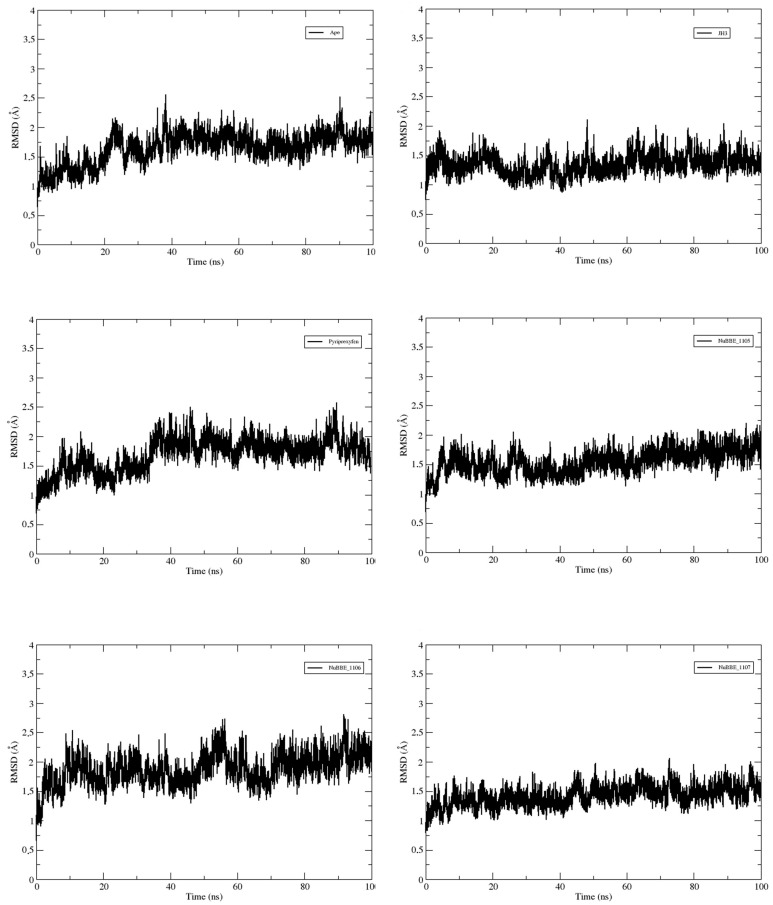
RMSD graph of the backbone of the Apo protein mJHBP, and mJHBP complexed with the ligands JH3, pyriproxyfen, NuBBE_1105, NuBBE_1106, and NuBBE_1107.

**Figure 4 molecules-28-06871-f004:**
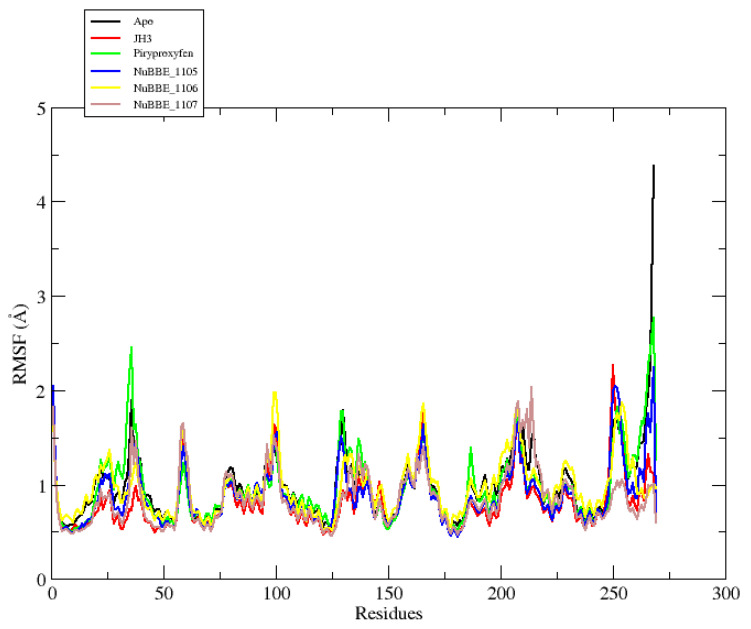
RMSF graph of the Apo protein mJHBP, and mJHBP complexed with JH3 ligands, pyriproxyfen, NuBBE_1105, NuBBE_1106, and NuBBE_1107.

**Figure 5 molecules-28-06871-f005:**
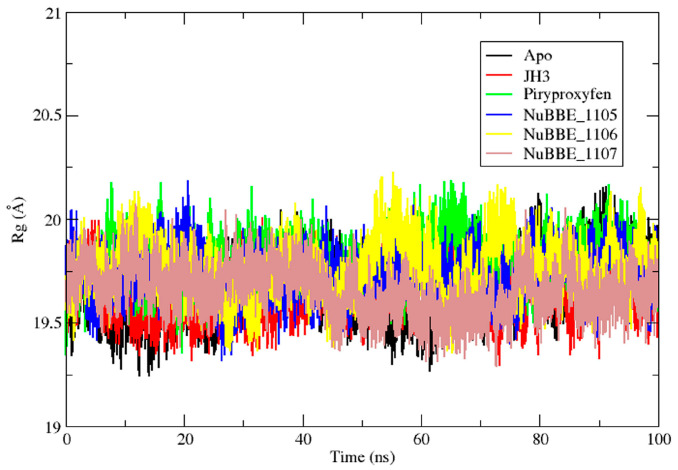
Radius of rotation (Rg) of the protein mJHBP Apo, and mJHBP complexed with the ligands JH3, pyriproxyfen, NuBBE_1105, NuBBE_1106, and NuBBE_1107.

**Figure 6 molecules-28-06871-f006:**
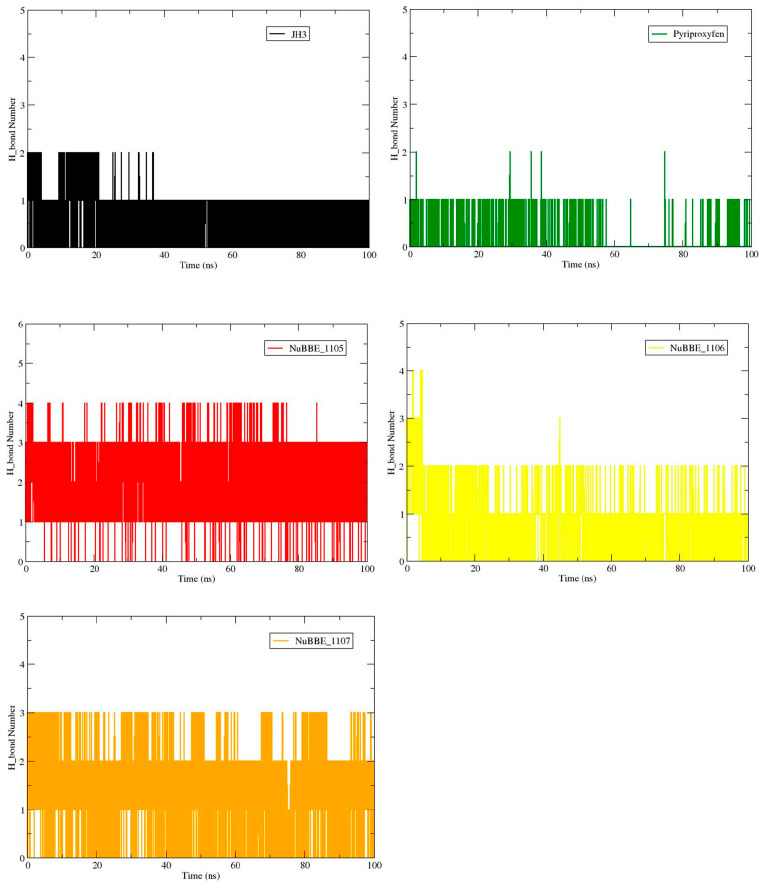
Number of hydrogen bonds between the ligands and the mJHBP protein during the 100 ns MD simulation time.

**Figure 7 molecules-28-06871-f007:**
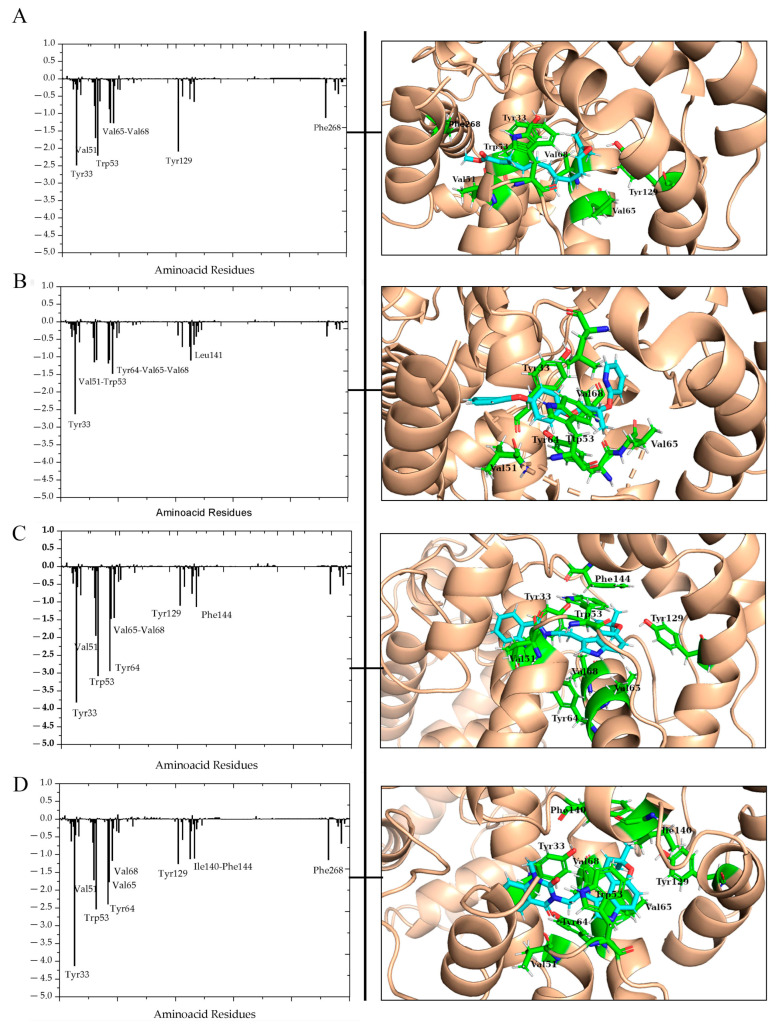
Per-residue binding free energy decomposition (kcal/mol) of mJHBP–JH3 (**A**), mJHBP–pyriproxyfen (**B**), mJHBP–NuBBE_1105 (**C**) and mJHBP–NuBBE_1107 (**D**) systems after 100 ns of MD simulation.

**Table 1 molecules-28-06871-t001:** Binding affinity energies for ligands complexed at the active site of the Aedes aegypti juvenile hormone binding protein (mJHBP, PDB ID: 5V13).

Ligands	PLPChem	Structures
NuBBE_1107 (3-(2-(7,7-dimethyl-3,7-dihydropyrano [3,2-e]indol-1-yl)ethyl-1-methylquinazoline-2,4(1H,3H)-dione)	113.05	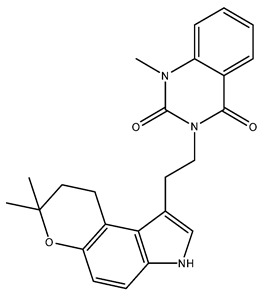
NuBBE_1105 (3-(2-(7,7-dimethyl-3,7-dihydropyrano[3,2-e]indol-1-yl)ethyl)quinazoline-2,4(1H,3H)-dione)	111.34	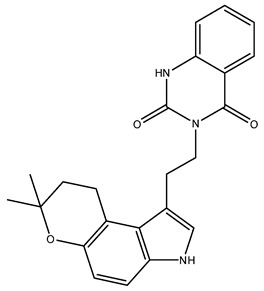
NuBBE_1106 (3-(2-(7,7-dimethyl-3,7-dihydropyrano[3,2-e] indol-1-yl)ethyl)-1-hydroxyquinazoline-2,4(1H,3H)-dione)	107.28	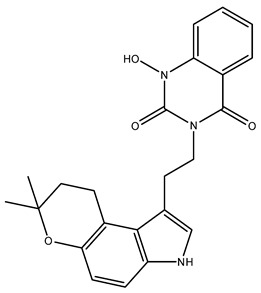
JH3 (2E,6E)-9-[(2R)-3,3-dimetiloxiran-2-il]-3,7-dimetilnona-2,6-dienoato	85.70	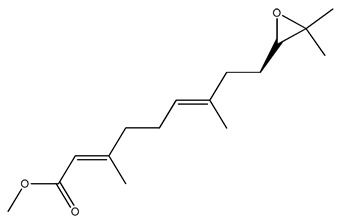
Pyriproxyfen 2-[1-(4-phenoxyphenoxy)propan-2-yloxy]pyridine	92.75	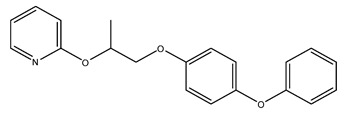

**Table 2 molecules-28-06871-t002:** MMGBSA binding free energies (∆*G*_bind_) and their components for the complexes under study. All values are reported in kcal·mol^−1^.

Ligand ID	ΔE_vdW_	ΔE_ele_	ΔG_GB_	ΔG_SASA_	ΔG_bind_
JH3	−46.55 ± 0.07	−8.96 ± 0.05	20.06 ± 0.01	−6.15 ± 0.03	−41.60 ± 0.07
Pyriproxyfen	−45.99 ± 0.07	−3.38 ± 0.07	19.98 ± 0.06	−6.36 ± 0.01	−35.75 ± 0.07
NuBBE_1105	−56.55 ± 0.08	−20.50 ± 0.10	34.71 ± 0.05	−6.49 ± 0.01	−48.84 ± 0.09
NuBBE_1106	−39.90 ± 0.08	−48.60 ± 0.19	81.33 ± 0.15	−4.38 ± 0.01	−11.55 ± 0.11
NuBBE_1107	−58.49 ± 0.07	−17.44 ± 0.09	32.96 ± 0.05	−6.73 ± 0.01	−49.70 ± 0.08

**Table 3 molecules-28-06871-t003:** Hydrogen bonds between the compounds JH3, pyriproxyfen, NuBBE_1105, NuBBE_1106, and NuBBE_1107 and mJHBP that had at least 2.0% occupancy throughout 100 ns of simulation time.

Complex	Hydrogen Bond Formation	Distance (Å)	Occupancy (%)
AagJHBP–H3	Tyr129@HH-LIG@O3	2.77	83.47
	Tyr148@HH-LIG@O2	2.77	9.26
AagJHBP–Pyproxyfen	Tyr64@HH-LIG@O	2.81	6.62
	Tyr133@HH-LIG@O1	2.81	2.41
AagJHBP–NuBBE_1105	Tyr64@HH- LIG@O2	2.82	95.93
	Tyr129@HH -LIG@O1	2.83	65.50
	Trp53@HE1-LIG@O	2.88	38.26
AagJHBP–NuBBE_1106	Tyr129@HH-LIG@O2	2.73	87.40
	Gly146@O-LIG@H4	2.59	5.39
AagJHBP–NuBBE_1107	Tyr64@HH-LIG@O	2.73	75.05
	Tyr129@HH-LIG@O1	2.83	46.13
	Trp53@HE1-LIG@O2	2.82	20.09

## Data Availability

Not applicable.

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
