# Peer review of "Exploring Natural Alkaloids from Brazilian Biodiversity as Potential Inhibitors of the Aedes aegypti Juvenile Hormone Enzyme: A Computational Approach for Vector Mosquito Control"

_molecules, 2023, doi:10.3390/molecules28196871_

Round 1

Reviewer 1 Report

In the present study, authors performed the computational modeling to screen alkaloids from Brazilian biodiversity as potential inhibitors against the Aedes aegypti juvenile hormone binding protein. Molecular docking and dynamics simulations identified two alkaloids with high predicted binding affinity, suggesting they could act as natural insecticides.

Overall, the work is well-written and provides valuable insights into the respective scientific arena. However, I have a concern regarding the use of the term "arboviruses" in the title, as it is not adequately discussed or connected with the main focus of the study. Also, the term "arboviruses" implies a group of viruses that are transmitted to humans and other animals through arthropod vectors such as mosquitoes and ticks and here in the present study, only A. aegypti has taken into consideration. So, it would be helpful if the authors clarify the rationale for choosing the current title and provide a stronger connection between the title and the main theme of the research. Additionally, if the authors decide to retain the term "arboviruses" in the title, they should ensure that the text provides sufficient background and context to justify its inclusion.

The introduction could provide more background just by restructuring the existing content with the focus on the importance and challenges of controlling A. aegypti specifically.

The authors should discuss limitations of their in silico-only approach and suggest future directions like in vitro binding assays, toxicity studies in mosquitoes, etc. to further validate the identified compounds.

There are some grammatical/language errors throughout that need to be edited by a native English speaker. For example, inconsistent capitalization, sentence fragments, etc.

To maintain consistency, all the nomenclatures should be written in italics, and the manuscript should be checked for similar typographical errors.

The figures could be improved for clarity - axis labels, larger text, consistent formatting. Make sure they are readable as standalone visuals (especially Fig. 2).

Reviewer 2 Report

Dear author,

I have reviewed the manuscript entitled "Exploring Natural Alkaloids from Brazilian Biodiversity as Potential Inhibitors of the Aedes aegypti Juvenile Hormone Enzyme: A Computational Study for the Control of the Main Vector of Important Arboviruses" finding a very interesting, valuable and necessary article that can serve as inspiration for future studies that allow the development of new insecticides that help combat diseases caused by Arbovirus vectors.

Allow me to suggest to you the following points:

1. The Ligand efficiency (LE) parameter is usually used as “a useful metric for lead selection” (10.1186/s13321-019-0330-2), in this case, it could be useful to better classify the best alkaloids. I suggest including it in your study.

2. Improve Table 1 to give the manuscript a better appearance.

3. The labels and labels of the amino acids in Figure 2 cannot be seen correctly. Authors are requested to increase the font size.

4. Figure 3 does not allow us to appreciate the individual behavior of the RMSD of the backbone of the Apo protein mJHBP and complexe, please improve the way of presenting the graph or separate the graphs into smaller groups to improve their appearance.

The other aspects of the manuscript are well done, a good job.

greetings
